# Enhancement of Sphingomyelinase-Induced Endothelial Nitric Oxide Synthase-Mediated Vasorelaxation in a Murine Model of Type 2 Diabetes

**DOI:** 10.3390/ijms24098375

**Published:** 2023-05-06

**Authors:** Éva Ruisanchez, Anna Janovicz, Rita Cecília Panta, Levente Kiss, Adrienn Párkányi, Zsuzsa Straky, Dávid Korda, Károly Liliom, Gábor Tigyi, Zoltán Benyó

**Affiliations:** 1Institute of Translational Medicine, Semmelweis University, H-1094 Budapest, Hungary; 2Eötvös Loránd Research Network and Semmelweis University (ELKH-SE) Cerebrovascular and Neurocognitive Disorders Research Group, H-1052 Budapest, Hungary; 3Department of Physiology, Semmelweis University, H-1094 Budapest, Hungary; 4Institute of Biophysics and Radiation Biology, Semmelweis University, H-1094 Budapest, Hungary; 5Department of Physiology, University of Tennessee Health Science Center, Memphis, TN 38163, USA

**Keywords:** sphingolipids, sphingomyelinase, vasorelaxation, endothelial nitric oxide synthase, type 2 diabetes, thromboxane prostanoid receptor

## Abstract

Sphingolipids are important biological mediators both in health and disease. We investigated the vascular effects of enhanced sphingomyelinase (SMase) activity in a mouse model of type 2 diabetes mellitus (T2DM) to gain an understanding of the signaling pathways involved. Myography was used to measure changes in the tone of the thoracic aorta after administration of 0.2 U/mL neutral SMase in the presence or absence of the thromboxane prostanoid (TP) receptor antagonist SQ 29,548 and the nitric oxide synthase (NOS) inhibitor L-NAME. In precontracted aortic segments of non-diabetic mice, SMase induced transient contraction and subsequent weak relaxation, whereas vessels of diabetic (*Lepr^db^*/*Lepr^db^*, referred to as db/db) mice showed marked relaxation. In the presence of the TP receptor antagonist, SMase induced enhanced relaxation in both groups, which was 3-fold stronger in the vessels of db/db mice as compared to controls and could not be abolished by ceramidase or sphingosine-kinase inhibitors. Co-administration of the NOS inhibitor L-NAME abolished vasorelaxation in both groups. Our results indicate dual vasoactive effects of SMase: TP-mediated vasoconstriction and NO-mediated vasorelaxation. Surprisingly, in spite of the general endothelial dysfunction in T2DM, the endothelial NOS-mediated vasorelaxant effect of SMase was markedly enhanced.

## 1. Introduction

Sphingolipids, derived from sphingomyelin metabolism, have been implicated as important mediators in the physiology and pathophysiology of the cardiovascular system [1,2,3,4,5,6]. Sphingomyelinase (SMase) catalyzes the conversion of sphingomyelin to ceramide, which is the precursor of other sphingolipid mediators, including ceramide-1-phosphate (C1P), sphingosine (Sph), and sphingosine-1-phosphate (S1P) [7]. The majority of S1P-induced biological effects are mediated by G-protein-coupled receptors (GPCRs), termed S1P_1–5_ [8]. Other sphingolipid mediators may exert biological effects by directly interacting with membrane or intracellular protein targets, independently of the activation of S1P receptors [5,9,10,11].

Based on the optimal pH for their catalytic activity, SMase isoforms can be divided into three groups: alkaline, acidic, and neutral [12]. The expression and known functions of alkaline SMases are mostly restricted to the gastrointestinal system, whereas acidic and neutral SMases are more widely expressed and involved in physiological and pathophysiological reactions in many systems, including the cardiovascular system. In the vasculature, SMases are implicated in the regulation of vascular tone and permeability as well as in causing atherosclerotic lesions and vascular wall remodeling [13]. Interestingly, neutral SMase has been reported to induce a wide range of changes in the vascular tone, depending on the species, vessel type, and experimental conditions (Table 1). Taken into account the large number of biologically active mediators (including ceramides, C1P, Sph, and S1P) that can be generated both extra- and intracellularly upon triggering the sphingolipid biosynthesis by neutral SMase, the diversity of vascular effects is not unexpected.

SMase enzymes are reportedly upregulated in certain cardiovascular and metabolic disorders, such as type 2 diabetes mellitus (T2DM) [13,26,27]. Sphingolipids have been implicated as important regulators of inflammatory processes in diabetes [28]. Stress conditions initiate changes in sphingolipid metabolism [29], and sphingolipids have emerged as key mediators of stress responses [30,31]. Extracellular stressors induce sphingolipid synthesis and turnover, thereby ‘remodeling’ sphingolipid profiles and their topological distribution within cells [32]. Emerging evidence not only demonstrates profound changes in sphingolipid pools and distribution under conditions of overnutrition [33,34,35], but also implicates sphingolipids in mediating cell-signaling responses that precipitate pathology associated with obesity [36]. In spite of the marked alterations in the metabolism and actions of sphingolipids in diabetes and recent observations indicating that ceramide may contribute to the development of diabetic endothelial dysfunction [37], relatively little is known about the effects of sphingolipids on vascular functions in T2DM. In the present study, we analyzed the effects of SMase on vascular tone under diabetic conditions in order to elucidate the signaling mechanisms involved.

## 2. Results

First, we verified the general metabolic and vascular phenotypes of the T2DM mice tested in the present study. Db/db mice reportedly develop obesity with elevated blood glucose levels and insulin resistance [38,39,40]. Accordingly, the body weight increased almost 2-fold (Figure 1A), whereas blood glucose levels increased 3-fold (Figure 1B) in db/db mice as compared to non-diabetic control littermates. Furthermore, the serum phosphorylcholine level was also significantly increased in the diabetic group (Figure 1C), which is consistent with the reported enhancement of SMase activity in type 2 diabetes [13,26,27]. According to literature data, acetylcholine (ACh)-evoked vasorelaxation of the aorta prepared from db/db mice is completely NOS-dependent [41], thus ACh was used to characterize the endothelial function. The vessels of db/db animals showed marked endothelial dysfunction, as indicated by the impairment of the dose-response relationship of ACh-induced vasorelaxation after precontraction with 10 μmol/L PE (Figure 1D). The E_max_ value decreased to 50.8 ± 2.0% in diabetic vessels as compared to controls (65.8 ± 3.9%). However, there was no significant difference in the EC_50_ values (34.7 ± 16.0 nM vs. 55.7 ± 15.7 nM), indicating unchanged potency in spite of the reduced efficacy of endogenous NO upon stimulation of endothelial NOS (eNOS) by ACh. In contrast, reactivity of the vascular smooth muscle to NO remained unaltered, as neither the E_max_ (105.2 ± 1.8% vs. 103.3 ± 2.2%) nor the EC_50_ (10.7 ± 1.3 nM vs. 14.1 ± 2.0 nM) values of sodium nitroprusside (SNP)-induced vasorelaxation differed in vessels of db/db animals as compared to controls (Figure 1E). Taken together, these results confirm the T2DM-like metabolic and vascular phenotypes in db/db mice and suggest the in vivo enhancement of SMase activity as well.

Next, we determined the effect of nSMase on the active tone of control and db/db vessels (Figure 2A). After 10 μmol/L phenylephrine (PE)-induced precontraction, 0.2 U/mL nSMase elicited additional contraction in control vessels that reached its maximum at 7.2 min before relaxing back to the pre-SMase level by the end of the 20-min observation period. In contrast, nSMase in db/db vessels elicited completely different responses. After a marked initial relaxation elicited by 0.2 U/mL nSMase during the first 5 min, the tone of the db/db vessels remained in a relaxed state below the level of the initial tension. From the shape of the tension curve, it appeared that in addition to the overriding relaxation, there was a delayed and transient constriction response with a time course similar to that observed in control vessels, but it was unable to overcome the robust dilatation. Evaluation of the AUC (Figure 2B) and the maximal changes in the vascular tone (Figure 2C) also supported the conclusion that there is a marked difference in the vascular effects of nSMase between control and db/db mice: contraction dominates in the former, whereas the latter is characterized by reduction of the vascular tone.

Our next aim was to differentiate the constrictor and relaxant components of the vascular tension changes in response to nSMase. In porcine coronary arteries [14] and in the carotid arteries of spontaneously hypertensive rats [23,24,25], prostanoids acting on TP receptors have been implicated in mediating the vasoconstrictor effect of SMase. Therefore, we hypothesized that thromboxane prostanoid (TP) receptors also mediate the nSMase-induced vasoconstriction in our murine aorta model. To test this hypothesis, the TP receptor antagonist SQ 29,548 was administered to the organ chambers 30 min prior to administration of nSMase. Blockade of TP receptors not only abolished the vasoconstriction but also converted it to a transient vasorelaxation in control vessels (Figure 3A). The maximum relaxation was reached at 5.5 min after the administration of nSMase, and the vascular tone returned to baseline after 10 min. TP receptor inhibition also markedly changed the vascular response to nSMase in the db/db group: the vasorelaxation was enhanced to more than 70% and reached its maximum at 6.5 min. After its peak, the relaxation decreased, but the vascular tone failed to return to the pre-SMase level even after 20 min. Both the AUC (Figure 3B) and the peak vasorelaxation (Figure 3C) values showed marked differences between the two experimental groups, indicating that the strongly enhanced and prolonged vasorelaxant capacity is responsible for the differences between the vasoactive effects of nSMase in db/db and control vessels. This finding was very surprising in light of the diminished ACh-induced vasorelaxation that we had observed in db/db animals (Figure 1D) and was not consistent with the large body of literature indicating diminished endothelium-dependent vasorelaxation in T2DM.

Next, we aimed to analyze the mechanism of the enhanced nSMase-induced vasorelaxation in the vessels of db/db mice. Theoretically, it could be due to the enhancement of eNOS-mediated vasorelaxation or to the onset of an NO-independent mechanism. To clarify this question, the vessels were incubated with the NOS inhibitor L-NAME (100 μM) in addition to the TP receptor blocker SQ 29,548 (1 µM) for 30 min prior to 0.2 U/mL nSMase administration. L-NAME at a concentration of 100 μM abolished the vasorelaxation observed in the presence of 1 µM SQ 29,548 both in control and in db/db vessels (Figure 4A). There were no significant differences between the two groups either in the AUC (Figure 4B) or in the maximal change of tension values (Figure 4C). These results indicate that the same secondary signaling pathways—namely TP receptors and eNOS—mediate the vasoactive effects of nSMase in health and in T2DM.

Finally, we aimed to investigate the possible involvement of downstream sphingolipid metabolites in the vasorelaxing effect of nSMase observed in db/db mice. Thus, aortic segments isolated from db/db were treated with either the ceramidase inhibitor D-erythro MAPP (10 µM) or SKI-II (1 µM), a sphingosine kinase inhibitor. In order to examine the vasorelaxant effects exclusively, vessels were pre-treated with 1 µM SQ 29,568. Neither MAPP nor SKI-II could affect the vasorelaxation evoked by 0.2 U/mL nSMase (Figure 5A). In addition, evaluation of AUC (Figure 5B) and maximal vasorelaxing responses (Figure 5C) showed no significant differences in MAPP or SKI-II-treated vessels compared to vehicle treated ones. These results suggest that the effect of nSMase is not mediated by sphingosine or sphingosine-1-phosphate.

## 3. Discussion

Findings of the present study indicate that nSMase-induced changes in vascular tension involve both vasoconstriction and vasorelaxation in murine vessels. Our results suggest that the former is mediated by the release of prostanoids and activation of TP receptors, whereas the latter is mediated by eNOS. Surprisingly, nSMase-induced eNOS-mediated vasorelaxation is markedly enhanced in the vessels of db/db mice in spite of the endothelial dysfunction indicated by the diminished vasorelaxation evoked by ACh. Therefore, nSMase appears to be able to induce enhanced NO release from endothelial cells in T2DM.

Vasoconstriction in response to SMase has been reported in a number of studies, although the mechanisms mediating this effect appear to be highly variable depending on the experimental conditions, including species, vascular region, and integrity of the endothelium (see Table 1). Release of prostanoids and consequent activation of TP receptors have been proposed in porcine coronary arteries [14] as well as in the carotid arteries of spontaneously hypertensive rats [23,24,25]. In our study, nSMase-induced contraction was found to be TP receptor-dependent in both control and db/db mice, indicating that nSMase stimulates the release of TXA_2_ from the aortic rings.

There might be at least three different sources for the SMase-induced arachidonic acid formation necessary for TXA_2_ production [42]. One such possibility is that diacylglycerol (DAG) would accumulate while sphingomyelin synthase converted the newly generated ceramide back to sphingomyelin, and DAG lipases would provide arachidonic acid for the production of TXA_2_ [43]. Another mechanism might relate to the observation that C1P can allosterically activate phospholipase A_2_ (PLA_2_) [44], which leads to arachidonic acid formation [45]. It might be important in this context that the gene encoding ceramide kinase (*CERK*) is upregulated in T2DM [46]. Finally, S1P has been reported recently to regulate prostanoid production in a S1P receptor-dependent manner [47]. It should be noted that mechanisms other than prostanoid release might also mediate the vasoconstrictor effect of SMase, i.e., modulation of different ion channels or initiating the Rho signaling pathway [18,19,22,48].

Vasorelaxation in response to nSMase appears to be endothelial NO-dependent, as L-NAME completely abolished the decrease in vascular tone in both control and db/db vessels. Without L-NAME, relaxation was dramatically increased in db/db-derived vascular rings. This is unexpected because endothelial dysfunction with consequential decreased vasorelaxant capacity is considered to be a hallmark of T2DM-like conditions. A potential explanation may be related to the altered structure of the plasma membrane in T2DM [49]. Normally, sphingomyelin (SM) represents about 10–20% of the lipids in the plasma membrane, mostly residing in the outer leaflet. However, most of these are found in the caveolae, and SMase is thought to be a regulator of lipid microdomains [50,51]. Pilarczyk and colleagues provided evidence that in db/db mice, the endothelial lining of the aorta contains 10-fold larger lipid raft areas enriched in SM as compared to controls [49]. This arrangement might be related to the decreased NO-release in T2DM, as eNOS is inhibited by caveolin-1 [52], which is considered to be an important regulator of eNOS [53,54,55]. In our experimental setting, nSMase-induced degradation of sphingomyelin could interfere with this caveolar structure and induce the detachment of eNOS from caveolin-1, leading to high amounts of NO released from the endothelium of db/db vascular rings. This hypothesis is supported by the observations of Mogami et al. [21], indicating that SMase causes endothelium-dependent vasorelaxation through Ca^2+^-independent endothelial NO production in bovine aortic valves and coronary arteries. They also reported SMase-induced translocation of endothelial NOS from plasma membrane caveolae to the intracellular region. Furthermore, protein expression levels of caveolin-1 were reported to be significantly higher in the aorta of db/db mice, and this was thought to be related to the impaired aortic relaxation of C57BL/KsJ mice [56]. In order to find out if downstream sphingolipid mediators such as sphingosine or sphingosine-1-phosphate contribute to the enhanced vasorelaxation seen in db/db aortic segments, we also investigated whether inhibition of ceramidase or sphingosine-kinase could interfere with the vasorelaxation. Neither D-erythro-MAPP nor SKI-II could diminish the nSMase-induced vasorelaxation in db/db vessels; therefore, it is unlikely that the release of these mediators participates in the effect. This finding strengthens our hypothesis that the enhanced relaxation in db/db vessels is mediated by a direct effect of nSMase on the membrane structure. On the other hand, we cannot rule out that the ceramide-related pathway might be involved in the SMase-induced contractions as well [18,23], as we did not address this question in our experiments. Finally, the potentially increased NO-sensitivity of guanylate cyclase (sGC) [57], which could be related to the dysfunctional NO-release observed in T2DM, should also be considered, as this would sensitize sGC to NO and result in enhanced NO-mediated vasorelaxation. However, this mechanism can be excluded in our present experiments, as the SNP dose-response curve remained unchanged in db/db vessels (Figure 1E), indicating that the sensitivity of the vascular smooth muscle to NO was not upregulated.

Sphingolipid metabolism is markedly altered in T2DM and related conditions [58,59,60,61,62], and the observed changes in endothelial lipid rafts [49] might be a consequence of the disrupted plasma membrane lipid metabolism. On the other hand, T2DM has several characteristics that resemble a chronic inflammatory disease [63]. Cytokines that accumulate in chronic inflammation, such as tumor necrosis factor alpha (TNF-α) and interleukin 1 beta (IL-1β), can also induce marked changes in sphingolipid metabolism [6,64,65]. Our observation that serum phosphorylcholine levels were increased in the db/db group might be a strong indicator of the altered in vivo sphingolipid metabolism in our animal model and agrees with the literature; however, it cannot be excluded that enzymes other than nSMase also contribute to this increase (e.g., other SMases, choline-kinase, phospholipase C, ENPP, etc.).

As a limitation of our study, it has to be mentioned that the characteristics of the pathophysiological conditions in the db/db mouse model differ from those of human T2DM in some aspects [66]. For example, db/db mice do not necessarily develop hypertension and may have high levels of high-density lipoprotein and a reduced tendency toward atherosclerosis [67]. Therefore, due to the more severe endothelial dysfunction, the enhancement of nSMase-induced eNOS-mediated vasorelaxation may be limited in humans with T2DM. A further limitation of our study is that we tested only one single dose of bacterial nSMase. This 0.2 U/mL dose of exogenously administered bacterial nSMase represents the upper range used in the literature [14,15,16,17,18,20,21,22,23], as our aim was to evaluate the consequences of a robust activation of sphingomyelinase degradation. Further studies may aim to elucidate the exact dose-response relationship for SMase-induced vasorelaxation and vasoconstriction in db/db mice or other T2DM-related conditions, which may also help to clarify the exact molecular mechanisms involved. Though bacterial and different mammalian nSMase orthologues share low overall homology, the ‘catalytic core’ residues are strongly conserved [68]. Therefore, it is likely that the addition of bacterial nSMase mimics the overactivation of mammalian nSMase and leads to similar cellular effects—in our case, the release of thromboxane and NO [69,70].

## 4. Materials and Methods

All procedures were carried out according to the guidelines of the Hungarian Law of Animal Protection (40/2013). The procedures were approved by the National Scientific Ethical Committee of Animal Experimentation (PE/EA/924-7/2021, accepted: 7 September 2021).

### 4.1. Animals and General Procedures

The BKS db diabetic mouse strain (JAX stock #000642) was obtained from the Jackson Laboratory (Bar Harbor, ME, USA) and has been maintained in our animal facility by mating repulsion double heterozygotes (*Dock7^m^* +/+ *Lepr^db^*). Littermate adult male diabetic (*Lepr^db^*/*Lepr^db^*, referred to as db/db) and misty (*Dock7^m^*/*Dock7^m^*, referred to as control) mice were selected for experiments. All mice investigated in this research were male and aged between 90 and 180 days. Animals were weighed, and blood samples were collected by cardiac puncture followed by transcardial perfusion with 10 mL of heparinized (10 IU/mL) Krebs solution under deep ether anesthesia, as described previously [71]. Nonfasting blood glucose was measured by a Dcont IDEÁL biosensor-type blood glucose meter (77 Elektronika Kft.; Budapest, Hungary). In some experiments, additional blood samples were collected, allowed to clot for 30 min at room temperature, and centrifuged at 2000× *g* for 15 min at 4 °C. Serum was snap frozen for a later phosphorylcholine assay, which was based on the method described by Hojjati and Jiang [72] using a commercially available kit (item No. 10009928, Cayman Chemical; Ann Arbor, MI, USA).

### 4.2. Myography

The thoracic aorta was removed and cleaned of fat and connective tissue under a dissection microscope (M3Z, Wild Heerbrugg AG; Gais, Switzerland) and immersed in a Krebs solution of the following composition (mmol/L): 119 NaCl, 4.7 KCl, 1.2 KH_2_PO_4_, 2.5 CaCl_2_·2 H_2_O, 1.2 MgSO_4_·7 H_2_O, 20 NaHCO_3_, 0.03 EDTA, and 10 glucose at room temperature and pH 7.4. Vessels were cut into ~3 mm-long segments and mounted on stainless steel vessel holders (200 µm in diameter) in a conventional myograph setup (610 M multiwire myograph system; Danish Myo Technology A/S; Aarhus, Denmark). Special care was taken to preserve the endothelium.

The wells of the myographs were filled with an 8 mL Krebs solution aerated with carbogen. The vessels were allowed a 30-min resting period, during which the bath solution was warmed to 37 °C and the passive tension was adjusted to 15 mN, which was determined to be optimal in a previous study [71]. Subsequently, the tissues were exposed to a 124 mmol/L K^+^ Krebs solution (made by isomolar replacement of Na^+^ by K^+^) for 1 min, followed by several washes with normal Krebs solution. Reactivity of the smooth muscle was tested by a contraction evoked by 10 μmol/L PE, and reactivity of the endothelium was tested by following the PE-evoked contraction with administration of 0.1 μmol/L ACh. After repeated washing, during which the vascular tension returned to the resting level, the segments were exposed to a 124 mmol/L K^+^ Krebs solution for 3 min in order to elicit a reference maximal contraction. Subsequently, after a 30-min washout, increasing concentrations of PE (0.1 nmol/L to 10 μmol/L) and ACh (1 nmol/L to 10 μmol/L) were administered to determine the reactivity of the smooth muscle and the endothelium, respectively. Following a 30-min resting period, the vessels were precontracted to 70–90% of the reference contraction by an appropriate concentration of PE, and after contraction had stabilized, the effects of 0.2 U/mL nSMase (SMase from *B. cereus*, Sigma-Aldrich; St. Louis, MO, USA) were investigated for 20 min. Bacterial SMase functions at neutral pH and is reportedly a useful tool for mimicking the biological effects of activation of cellular SMase [73,74]. In some experiments, the selective TP receptor antagonist SQ 29,548 (1 µM) with or without the nitric oxide synthase (NOS) inhibitor L-NAME (100 µM) was applied to the baths 30 min prior to administration of nSMase. In order to block ceramidase or sphingosine-kinase enzymes, D-erythro MAPP (10 µM) or SKI-II (1 µM) were used, respectively. Finally, to test the sensitivity of the smooth muscle to NO, SNP (0.1 nmol/L to 10 μmol/L) was administered after a stable precontraction elicited by 1 μmol/L PE.

### 4.3. Data Analysis

An MP100 system and AcqKnowledge 3.72 software from Biopac System Inc. (Goleta, CA, USA) were used to record and analyze changes in the vascular tone. All data are presented as mean ± SE, and “*n*” indicates the number of vascular segments tested in myography experiments or the number of animals tested in the case of body weight, blood glucose, and serum phosphorylcholine levels. Maximal changes in the vascular tone were calculated as a percentage of precontraction. To evaluate the temporal pattern of nSMase-induced vasoactive responses, individual curves were constructed and averaged, showing the changes in vascular tone for 20 min after the application of nSMase. Area under the curve (AUC) values were calculated from individual experiments for quantification of the overall vasoactive effect. The statistical analysis was performed using the GraphPad Prism software v.6.07 from GraphPad Software Inc. (La Jolla, CA, USA). Student’s unpaired *t*-test was applied when comparing two variables, and a *p* value of less than 0.05 was considered to be statistically significant. The effects of cumulative doses of PE and ACh were evaluated by dose-response curve fitting for the determination of E_max_ and EC_50_ values. All data presented in this study are available in the Appendix A.

### 4.4. Reagents

All reagents in this study, including nSMase, were purchased from Sigma-Aldrich (St. Louis, MO, USA), except SQ 29,548, which was from Santa Cruz Biotechnology (Dallas, TX, USA).

## 5. Conclusions

Administration of nSMase induces TP receptor-mediated vasoconstriction and eNOS-mediated vasorelaxation in murine vessels. In spite of endothelial dysfunction in db/db mice, the vasorelaxant effect of nSMase is markedly augmented. SMase-mediated disruption of SM in endothelial lipid rafts might represent a possible mechanism responsible for enhanced NO generation in T2DM. An intriguing interpretation of our finding is that the retraction of eNOS in sphingomyelin-rich microdomains of the endothelial plasma membrane could contribute significantly to the development of vascular dysfunction in T2DM.

## Figures and Tables

**Figure 1 ijms-24-08375-f001:**
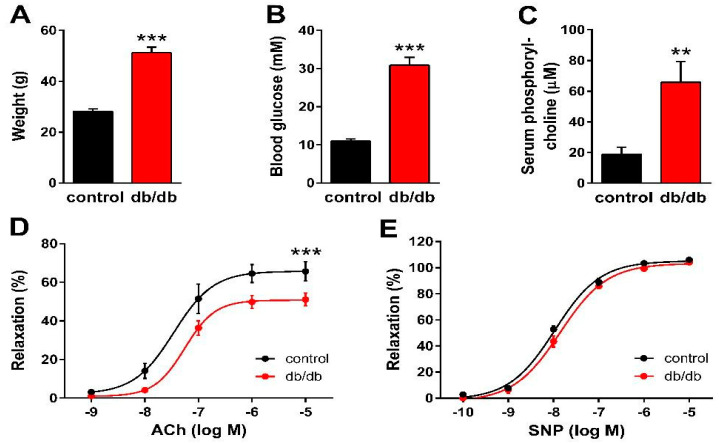
Manifestation of the metabolic and vascular phenotypes of T2DM in db/db mice. Body weight (**A**), as well as non-fasting blood glucose (**B**) and serum phosphorylcholine levels (**C**), increased in db/db mice as compared to controls (** *p* < 0.01, *** *p* < 0.001 vs. control group; Student’s unpaired *t*-test, *n* = 13–22). ACh-induced relaxation diminished (**D**), while the reactivity of the vascular smooth muscle to sodium nitroprusside (SNP) remained unaltered (**E**) in vessels of db/db mice as compared to controls (mean ± SEM, *** *p* < 0.001 vs. control; dose-response curve fitted to *n* = 12–24).

**Figure 2 ijms-24-08375-f002:**
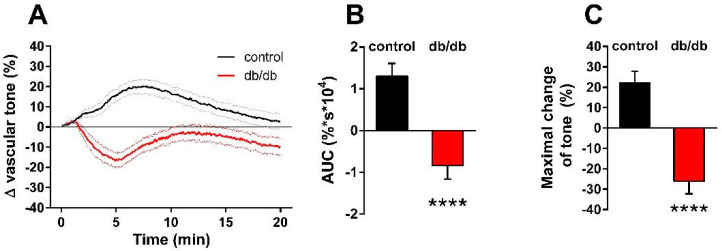
Effects of nSMase on the vascular tone. Application of 0.2 U/mL nSMase evoked a complex vascular effect with dominant contraction in control vessels and a more pronounced relaxation in vessels of db/db mice. Black and red lines on panel (**A**) represent average changes in tension of PE-precontracted vessels in control and db/db mice, respectively (dotted lines represent SEM). Both area under curve values (**B**) and maximal tension changes (**C**) were significantly different in vessels from db/db animals as compared to controls (mean ± SEM, Student’s unpaired *t*-test, **** *p* < 0.0001 vs. control; *n* = 51–49).

**Figure 3 ijms-24-08375-f003:**
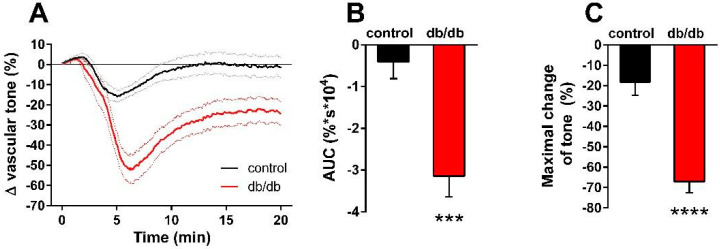
Effects of TP receptor blockade on nSMase-induced changes in the vascular tone. After inhibition of the TP receptor by 1 μM SQ 29,548, 0.2 U/mL nSMase relaxed both db/db and control vessels, with a significantly higher relaxation in the db/db group (**A**). Black and red lines in panel A represent average tension changes in PE-precontracted vessels of control and db/db mice, respectively, whereas dotted lines represent SEM. Both area under curve values (**B**) and maximal tension changes (**C**) were significantly different in vessels from db/db animals as compared to controls (mean ± SEM, Student’s unpaired *t*-test, *** *p* < 0.001 vs. control; **** *p* < 0.0001 vs. control; *n* = 20).

**Figure 4 ijms-24-08375-f004:**
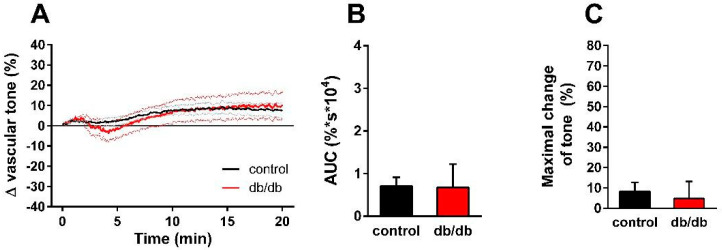
Effects of combined TP receptor and NOS blockade on nSMase-induced changes in vascular tone. After incubation of the vessels with 1 μM SQ 29,548 and 100 μM L-NAME for 30 min, 0.2 U/mL nSMase could no longer evoke a tension change in the thoracic aorta of control or db/db mice (**A**). Black and red lines in panel A represent average tension changes in PE-precontracted vessels of control and db/db mice, respectively (dotted lines represent SEM). Area under curve values (**B**) and maximal tension changes (**C**) were not different in vessels from db/db animals as compared to controls (mean ± SEM, *n* = 9–17).

**Figure 5 ijms-24-08375-f005:**
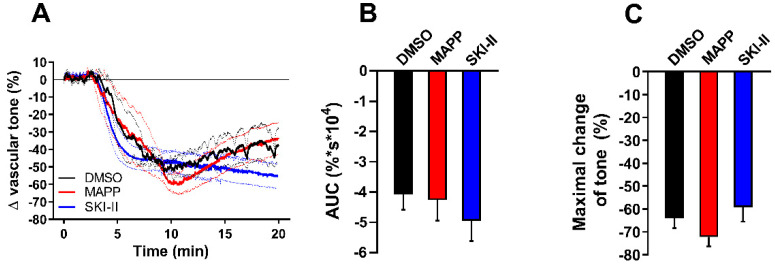
Effects of ceramidase and sphingosine-kinase inhibitors on nSMase-induced vascular tone changes in db/db mice. After incubating with 1 μM SQ 29,548 for 30 min, db/db vessels were treated with 10 μM D-erythro MAPP or 1 μM SKI-II. Neither MAPP nor SKI-II could affect the vasorelaxation induced by 0.2 U/mL nSMase (**A**). Black, red, and blue lines in panel A represent average tension changes in PE-precontracted vessels of vehicles, MAPP, and SKI-II treated vessels, respectively (dotted lines represent SEM). Area under curve values (**B**) and maximal tension changes (**C**) were not different in MAPP- or SKI-II-treated vessels as compared to vehicle controls (mean ± SEM, *n* = 4–5).

**Table 1 ijms-24-08375-t001:** Reported vasoactive effects of neutral SMase.

Species	Vessel	Vasoactive Effects	Proposed Mechanism	Refs.
*Yorkshire pig*	Coronary artery	Transient endothelium- dependent contraction followed by endothelium- dependent relaxation	Vasoconstriction: prostanoid(s) Vasorelaxation: NO	[14]
*Sprague*- *Dawley rat*	Thoracic aorta	Endothelium-independent relaxation	Inhibition of protein kinase C (PKC)	[15,16]
*Wistar rat*	Thoracic aorta	Partly endothelium-independent relaxation	Endothelium-mediated components are independent of NO or prostanoids Non-endothelial components are independent of PKC	[17]
*Mongrel dog*	Basilar artery	Endothelium-independent contraction	Activation of VDCC and PKC	[18]
*Wistar rat*	Pial venule (60–70 μm in diameter)	Constriction and spasm	Activation of VDCC, PKC, and MAP kinase	[19]
*Wistar rat*	Thoracic aorta	Endothelium-independent relaxation	Inhibition of both Ca^2+^-dependent and Ca^2+^- independent (RhoA-/Rho kinase-mediated) contractile pathways	[20]
*Cow*	Coronary artery	Endothelium-dependent relaxation	Ca^2+^-independent eNOS activation, involving phosphorylation on serine 1179 and dissociation of eNOS from plasma membrane caveolae	[21]
*Wistar rat*	Pulmonary artery	Endothelium-independent contraction	Activation of VDCC, PKCζ, and Rho kinase	[22]
*Wistar-Kyoto* (WKY) and *Spontaneously hypertensive rat* (SHR)	Carotid artery	SHR: strong endothelium- dependent contraction WKY: weak endothelium- dependent contraction	Vasoconstriction is mediated by PLA_2_- and COX2-mediated TXA_2_ release and attenuated by NO	[23,24,25]

NO, nitric oxide; PKC, protein kinase C; VDCC, voltage-gated calcium channel; MAP, mitogen activated protein; eNOS, endothelial NO synthase; PLA2, phospholipase A2; TXA2, thromboxane A2; COX, cyclooxygenase.

## Data Availability

The data presented in this study are available in the Appendix A.

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
