# Peer review of "Enhancement of Sphingomyelinase-Induced Endothelial Nitric Oxide Synthase-Mediated Vasorelaxation in a Murine Model of Type 2 Diabetes"

_ijms, 2023, doi:10.3390/ijms24098375_

Round 1
Reviewer 1 Report
In this study, the authors report on a dual effect of neutral sphingomyelinase on vascular tone in isolated vessels from mice. nSMase application to control mouse vessels caused a vasoconstriction that was reverted by a TP-R antagonist, while in vessels of diabetic db/db mice, nSMase caused a strong vasorelaxation through enhanced eNOS activation. The authors hypothesized that the enhanced eNOS in lipid rafts of the plasma membrane of endothelial cells contributes to the vascular dysfunction in type 2 diabetes.
Major points :
1- A main issue of this study is that the authors exclusively used a bacterial nSMase for all their experiments. It is not clear whether mouse nSMase (either Smpd2,3, or 4) or aSMase (Smpd1) have the same activity on vascular tone as the bacterial enzyme. Therefore, the authors must test at least in a key experiment one of the mouse enzymes.
2-In Fig. 1C, the authors show increased phosphorylcholine in serum of db/db mice arguing that under diabetic conditions, indeed a SMase is activated. Since phosphorylcholine is not only generated by nSMase, but also by aSMase and eventually a PC-PLC, it is no reliable measure of nSMase activity. Instead, the authors should an increased level of ceramides, for example by mass spectrometric quantifications of serum or even isolated vessels. Additionally, it would be of great interest to show not only ceramides, but also the downstream molecules sphingosine and sphingosine-1-phosphate. This would give a more clear picture of which molecules accumulate in diabetes and may mediate the vasorelaxing effect.
3-Can the effect on vascular tone seen in db/db mouse vessels be reversed by a commercial nSMase or aSMase inhibitor?
4-To ennarrow the sphingolipid involved, it would also be interesting to see whether nSMase in the presence of either ceramide inhibitors or sphingosine kinase inhibitors abrogates the vasorelaxing effect.
Reviewer 2 Report
In this paper, the authors showed that an administration of nSMase induced TP receptor-mediated vasoconstriction and eNOS-mediated vasorelaxation simultaneously in mice aorta. Moreover, the vasorelaxant effect of nSMase was markedly augmented in db/db mice aorta possibly by nSMase-induced translocation of eNOS from plasma membrane caveolae to the intracellular region. The authors findings and hypothesis are very interesting and such a mechanism might be a possible reason behind the inconsistent results of previous literature regarding NO-mediated vasorelaxation during diabetes. I encourage the authors to further develop the idea of nSMase's role in regulating endothelial function during diabetes, including those from small arteries. Overall, this is an interesting paper that deals with an important topic; however, this reviewer has a small number of concerns that should be addressed.
Major comments
1. Figure 1 D: As it stands, it is unclear what are the underlying mechanisms of impaired ACh-induced vasorelaxation in db/db mice. The authors need to provide the vasorelaxation data in the presence of a TP receptor antagonist as well as an eNOS blocker.
2. Discussion, lines 227-238: This hypothesis is very fascinating, but I was wondering how does eNOS dissociated from caveolin-1 lead to the generation of NO independently of intracellular Ca2+ rise. Could the authors elaborate more on this point?
3. I think it would be better to combine the data presented in Figure 2, 3, and 4 in a single Figure.
Minor comments
1. Material and Methods: The sex and age of the mice should be reported.
2. Introduction, line76, Ref 37: This paper reviews mainly focusing on the effect of ceramide on NO-mediated vasorelaxation. Because ceramide also alters vascular tone through inhibiting vascular ion channels (see Goto & Kitazno, Endocrine and Metabolic Science, 2020), I would suggest the authors to briefly add this information in the text.
3. It would be helpful for the readers to provide a schematic diagram, which explain the underlying mechanisms of vasoconstriction and vasorelaxation following nSMase administration in both control and db/db mice.
Round 2
Reviewer 1 Report
The authors have satisfactorily dealt with most of the concerns. New experiments have been included to strengthen the conclusion.
Reviewer 2 Report
The authors have adequately addressed all issues raised by this reviewer.